# Healthcare professionals' perspectives on barriers and facilitators to implementing a warning signs intervention for older rural-dwelling medical patients at risk for hospital readmission

**Mary T. Fox**[1,2]*, **Jeffrey I. Butler**[1,2], **Adam M. B. Day**[3], **Evelyne Durocher**[4], **Sherry Dahlke**[5], **Mark W. Skinner**[6], **Behdin Nowrouzi-Kia**[7], **Janet Yamada**[8], **Ilo-Katryn Maimets**[9]

1 School of Nursing, York University, Toronto, Ontario, Canada, 2 York University Centre for Aging Research and Education, Toronto, Ontario, Canada, 3 Northern Ontario School of Medicine, Thunder Bay, Ontario, Canada, 4 School of Rehabilitation Science, McMaster University, Hamilton, Ontario, Canada, 5 Faculty of Nursing, University of Alberta, Edmonton, Alberta, Canada, 6 Trent School of the Environment, Trent University, Peterborough, Ontario, Canada, 7 Department of Occupational Science and Occupational Therapy, University of Toronto, Toronto, Ontario, Canada, 8 School of Nursing, Toronto Metropolitan University, Toronto, Ontario, Canada, 9 Steacie Science and Engineering Library, York University, Toronto, Ontario, Canada

* maryfox@yorku.ca

## Abstract

### Introduction

Prior research has identified that older rural patients and their families view preparation for detecting and responding to worsening health after a hospital stay as their most pressing unmet need, and perceive an evidence-based warning signs intervention that prepares them to do so as highly likely to meet this need. Yet, little is known about healthcare professionals' perspectives about potential barriers and facilitators to implementing warning signs interventions, especially in rural communities.

### Aim

This study aimed to identify potential barriers and facilitators to healthcare professionals' provision of a warning signs intervention in rural communities.

### Materials and methods

In this qualitative descriptive study, we examined healthcare professionals' perspectives on potential barriers and facilitators to providing a warning signs intervention. A purposive, criterion-based sample of healthcare professionals, stratified by professional designation (three strata – nurses, physicians, and allied healthcare professionals) who provide health care to rural dwellers in Ontario, Canada participated

**Data availability statement:** In accordance with the guidelines set by the Health Sciences North Research Institute's Research Ethics Board, the authors are not permitted to make the study data publicly available. Due to the nature of the sample, which was recruited from small towns, there is a risk that participants' characteristics could be identifiable even in de-identified datasets. Consequently, the authors are unable to share the data publicly. However, researchers who are interested in accessing the data may submit a request to the hospital REB for consideration. Inquiries should be directed to the following contact: Research Ethics Board, Health Sciences North Research Institute, 1 Ramsey Lake Road, Sudbury, Ontario, P3E 5J1; Phone: 705-523-7100; Email: reb@hsnsudbury.ca.

**Funding:** This work is supported by the Canadian Institutes of Health Research, Funding Reference Number 163072, awarded to M.T.F. The funder had no role in the design, data collection and analysis, decision to publish, or preparation of the manuscript.

**Competing interests:** The authors have declared that no competing interests exist.

in semi-structured telephone focus-group discussions or 1:1 interviews on barriers and facilitators to delivering the intervention. Data were analyzed using conventional qualitative content analysis.

## Results

Twenty-seven healthcare professionals participated in focus groups and 15 in 1:1 interviews for a total of 42 healthcare professionals. Analysis by healthcare professional stratum revealed nine categories of barriers and facilitators: material resources; human resources; healthcare professional communication; healthcare professional knowledge and skill; healthcare professional buy-in; context of rural practice; patient- and family-specific characteristics; risks and liabilities; and timing of intervention delivery. Seven of these categories converged across healthcare professional strata. However, the reasons why different healthcare professional strata perceived the categories as important, and the ways in which they saw them functioning as barriers and facilitators, varied. Our findings shed light on barriers and facilitators that should be considered to ensure successful implementation of the intervention in rural communities.

## Discussion

This study adds to the limited research on rural healthcare professionals' perspectives on barriers and facilitators to delivering a warning signs intervention.

## Introduction

Hospital-to-home transitional care (TC) refers to healthcare services provided to patients to ensure their care needs continue to be met after discharge home. In response to high rates of hospital readmissions of older patients with complex health conditions, TC has become a health systems priority in many jurisdictions [1,2]. Nowhere is this more a priority than in rural communities. Compared to urban communities, rural communities have greater proportions of older people (aged 60+) [3,4] and people living with multiple chronic conditions (≥ 2 concurrent chronic conditions)[4,5], which are risk factors for hospital readmission [6,7]. Indeed, rural dwellers have significantly higher rates of emergency department visits and hospital readmissions during the 30-day post-discharge period, and up to 59% of their hospital readmissions are considered avoidable [8].

TC commences in hospital and is usually provided by nurses in conjunction with the interprofessional team [9]. TC models typically emphasize the importance of adequately preparing patients to recognize and respond appropriately to the signs that their health may be worsening [9,10]. A systematic review of TC trials concluded that a warning signs intervention, as part of multicomponent TC programs, is associated with decreased visits to the emergency department and fewer readmissions to the hospital [9].

Based on a literature synthesis, we devised a warning signs intervention aimed at helping patients and their families recognize and respond to the signs of worsening health [11]. The intervention begins with an in-hospital assessment of patients' and families' health literacy, their knowledge of the signs of worsening health relevant to the patient's condition(s), their capacity to recognize and respond to the signs at home, and their learning needs [11]. The intervention is delivered both prior to and after hospital discharge. To promote its adoption, the intervention includes educational strategies such as the 'teach-back' method and written materials with symbols or pictograms to facilitate comprehension of how to monitor health and identify and respond to the signs of worsening health, such as which signs constitute a medical emergency and the need to seek immediate medical assistance [11].

Older rural medical patients at risk for hospital readmission and their families have highlighted that knowing how to detect and respond to the signs of worsening health is their most pressing unmet TC need [12,13]. They emphasized having received little preparation on how to recognize and respond to these signs, and needing more preparation in this regard. These findings parallel other studies indicating that 41% of patients discharged from hospital are unaware of the signs that their health may be declining [14]. Medical patients who do not know what signs to monitor after hospital discharge are almost 3.5 times more likely to visit the emergency department or be readmitted to hospital compared to those who know what signs to monitor [15]. Given the global trend toward shorter hospital stays and early discharges, older medical patients are likely to be discharged when their acute conditions are not fully resolved [16]. Consequently, there is an urgent need to better prepare such patients and their families for recognizing and responding to signs of worsening health to prevent adverse health consequences that require hospital readmission. Yet, little is known about healthcare professionals' (HCP) perspectives on potential barriers and facilitators to implementing a warning signs intervention.

Previous research has examined barriers to TC in general (e.g., lack of expertise [17], fractured communication [18]) as well as facilitators (e.g., use of a pre-discharge check list, [17,18] strong interprofessional collaboration [18]). However, prior research has not explored HCPs' perspectives on barriers and facilitators to implementing a warning signs intervention, particularly in rural communities where such interventions are urgently needed to prevent avoidable patient deterioration in health and unnecessary hospital readmissions. Building knowledge in this area is essential to knowing how to support HCPs in delivering the intervention, thereby optimizing TC in rural communities. Consequently, this study aimed to identify HCPs' perspectives on barriers and facilitators to providing a warning signs intervention proposed for rural TC with older rural medical patients at risk for hospital readmission and their families.

## Materials and methods

### Design

This qualitative descriptive study was part of a larger multi-method study focused on gathering HCPs' perspectives on a warning signs intervention proposed for rural TC [11]. The study design was guided by the *intervention acceptability* [19] and *knowledge-to-action* [20] frameworks. Both frameworks are rooted in community-based, participatory research and emphasize the importance of engaging HCPs when examining intervention acceptability and implementation [19,20]. The frameworks guide researchers in systematically assessing HCPs' perspectives on the acceptability of an intervention, as well as assessing barriers and facilitators to its delivery, in order to design a plan [21] that aids HCPs' implementation of the intervention with fidelity to maximize its effectiveness [19,20]. This study focused on understanding HCPs' perspectives on barriers and facilitators to implementing the warning signs intervention.

### Ethical considerations

Ethics approval was obtained from the Research Ethics Board at York University (certificate #: e219-241) and from the Research Ethics Office at Health Sciences North (Project# 19–020). All participants provided written informed consent. The consent process entailed HCPs contacting the Research Associate (RA) who provided them with the consent form

and the opportunity to meet by telephone with him to review the study and what participation would entail. Participants were also offered the opportunity to ask questions vie telephone and/or email. All consenting participants sent their signed consent form to the RA electronically.

## Setting and sample

Our purposive, criterion-based sample of HCPs was stratified by professional designation. The three strata were: nurses (including nurse practitioners), physicians, and allied HCPs. Participants were recruited from rural and rural-serving juris-dictions of Ontario, initially from Southwestern and Northeastern Ontario and then from other rural areas in the province after recruitment slowed during the onset of the COVID-19 pandemic. Rural dwelling older adults in Ontario face chal-lenges such as limited access to care due to shortages of HCPs [22,23]. Such challenges are common in rural areas across North America [24]. Rural Ontario thus constitutes an ideal research recruitment setting.

The criteria for inclusion were that HCPs were: working ≥ 21 hours/week [25] in an Ontario hospital and/or community care (e.g., primary care or home care) setting and providing TC to rural patients with medical conditions. HCPs who had not participated in the larger study's survey on acceptability of the warning signs intervention [26] were ineligible to participate in this current study. HCPs within each stratum were invited to participate in separate focus group interviews. Given that some HCPs' work straddled hospital and community care settings, the focus groups included HCPs working in both settings.

Sampling for focus groups and interviews was conducted until it was determined that informational saturation was reached (i.e., when further data collection became redundant) [27]. A total of 46 HCPs expressed interest in participating in a focus group or 1:1 interview but two declined participation due to scheduling conflicts and three did not respond to follow-up requests. In total, 27 survey respondents participated in focus groups and 15 survey respondents participated in 1:1 interviews for a total sample size of 42. Five focus groups and three interviews were conducted with nurses, three focus groups and five interviews were conducted with allied HCPs, and eight 1:1 interviews were conducted with physi-cians. In total, eight focus groups and sixteen 1:1 interviews were conducted.

## Data collection

Recruitment was initiated by the research team knowledge-users who introduced the study at staff meetings, posted flyers at the hospital sites, and raised awareness of the study via email and social media platforms. Strategies to promote participation included holding the focus groups during non-working hours via telephone, providing a $75 gift card, and asking HCPs to refer colleagues [28]. A qualitative semi-structured interview guide, which had been pilot tested, was used to direct the focus group discussions and 1:1 interviews to further explore HCPs' perspectives on barriers and facilitators to providing the warning signs intervention to older rural medical patients at risk for hospital readmission and their families [11]. Questions prompted HCPs to discuss barriers and facilitators to delivering the warning signs intervention. Prior to the focus group discussions or interviews, participants were invited to complete an online demographic questionnaire (e.g., age, gender) and professional profile (e.g., years of experience, highest level of education).

Consenting HCPs were invited to participate in telephone focus groups (four to six HCPs per group) [29] or, if they were unable to attend due to scheduling conflicts, to participate in a 1:1 telephone interview. Participating HCPs were emailed the warning signs intervention logic model [11], which was synthesized from the empirical literature and our prior research [11,30–36], and described the intervention's goals, activities, mode of delivery, dose, anticipated benefits, and the human and material resources required to provide it.

Telephone focus groups and 1:1 interviews were conducted by a Research Associate with doctoral preparation in qualitative methods and experience in conducting telephone focus groups and interviews. Focus groups and interviews were audio-recorded, transcribed verbatim and were approximately 60 minutes in duration. Data were collected between 01-08-2019 and 31-10-2021.

## Data analysis

Descriptive statistics, in accordance with each variable's level of measurement, were used to describe HCPs' demographic and professional characteristics (e.g., mean for age, frequencies for gender). Conventional qualitative content analysis of the interview data, as described by Hsieh and Shannon [37], was performed concurrently with data collection to elucidate potential barriers and facilitators to delivering the intervention. Conventional qualitative content analysis is typically employed in studies describing a phenomenon (in our case, barriers and facilitators to implementing the warning signs intervention), and is most appropriate when research literature on that phenomenon is limited [37]. The key advantage of the conventional approach to qualitative content analysis is describing social phenomena in participants' own terms. Researchers using this approach immerse themselves in the data, while avoiding using preconceived categories, to generate categories rooted directly in the data.

Coding was facilitated by NVivo 12. We performed first- and second-cycle coding by applying codes to segments of text, and collating the quotes to support reporting. In line with Hsieh and Shannon's approach, coding categories were derived inductively from the interview transcripts. Analysis involved developing preliminary codes and organizing them into hierarchical categories. Each code and category were defined, the interconnections between them documented, and exemplar quotes for each category were selected. Coding was conducted independently by two members of the research team (MF and JB). The coding process involved each coder developing a list of potential codes on their own which they then compared for divergence and overlap. We identified barriers and facilitators that were common across all HCP strata, while also acknowledging negative cases which may speak to differential perspectives and experiences among HCP strata. Any discrepancies were reviewed and discussed between the coders. When there was disagreement amongst the coders as to the salience of a code, it was brought forward to the team and debated until consensus was achieved [38]. Codes for which we had consensus were integrated into the final codebook, and related codes were collapsed to create our final categories.

The data were then examined for patterns in HCPs' narratives based on their strata using role-ordered and conceptually-clustered matrices and analytic memoing. Specifically, we mapped the perspectives of each stratum of HCPs on the barriers and facilitators using matrices, which were summarized within each stratum, and within and across settings (hospital and community). Analytic memos were used to record our thoughts, interpretations, and reflections about the data. Strategies to support trustworthiness of the data were employed throughout the research process. Confirmability was increased through the creation of an audit trail, and credibility by having two research team members independently analyze the data and involving knowledge-users in interpreting the findings.[39] Dependability was promoted through detailed methodological reporting and transferability through the description of the sample's demographic features (e.g., gender, region of practice) [39].

## Results

Analysis by HCP strata revealed nine categories of barriers and facilitators: material resources; human resources; HCP communication; HCP knowledge & skill; HCP buy-in; context of rural practice; patient- and family-specific characteristics; HCP risks and liabilities; and timing of intervention delivery. Seven categories converged across HCP strata (see Table 1). However, the reasons why different HCP strata perceived the categories as important, and the ways in which they saw them functioning as barriers and facilitators, sometimes varied. Four categories of barriers and facilitators were only identified by two HCP strata. Below, we examine each category in turn. Individual HCP characteristics are outlined in Table 2, and illustrative quotes are presented in Table 3. We indicate where findings are specific to one HCP stratum; references to "participants" or "HCPs" denote all HCP strata.

### Barriers and facilitators identified by all HCP strata

**Material resources.** Participants conveyed that a lack of educational materials would prevent HCPs from providing the intervention. They foresaw insufficient condition-specific (e.g., Congestive Heart Failure, Chronic Obstructive Pulmonary

**Table 1. Barriers and Facilitators to Implementing the Warning Signs Intervention by Healthcare Professional Stratum.**

| Barriers/Facilitators | Nurses | Physicians | Allied Health |
|---|---|---|---|
| Material Resources | ✓ | ✓ | ✓ |
| Human Resources | ✓ | ✓ | ✓ |
| HCP Communication | ✓ | ✓ | ✓ |
| HCP Knowledge & Skill | ✓ | ✓ | ✓ |
| HCP Buy-In | ✓ | ✓ | ✓ |
| Context of Rural Practice | ✓ | ✓ | ✓ |
| Patient- and Family-Specific Characteristics | ✓ | ✓ | ✓ |
| HCP Risks & Liabilities | – | ✓ | ✓ |
| Timing of Intervention Delivery | ✓ | ✓ | – |

Note: HCP = Healthcare professional.

Disease) teaching materials as likely barriers. Participants, for example, explained that they were unsure of where to locate appropriate resources and many were only permitted to use employer-approved teaching materials, yet these were very limited. They identified that concise, pre-packaged written teaching materials for the medical conditions they most frequently encounter would facilitate their ability to provide the intervention. Participants described that these materials should be easily accessible in hard copy and available in different languages, so that patients could read them prior to discharge and take them home to review and share with their primary or home care professional. Participants also suggested that these materials contain colour-coded decision supports that designate urgency (i.e., green means no action is needed, yellow means watchful waiting is required, and red signals the need to seek urgent, immediate medical help such as calling an ambulance) and that written materials could be supplemented with instructional videos that patients could begin viewing while in hospital.

**Human resources.** Participants identified insufficient staff as an expected impediment to implementing the warning signs intervention and stressed that ongoing staffing shortages would not allow them sufficient time to deliver the intervention. Many recognized the intervention's value, but anticipated that it would add to their workloads. Consequently, participants indicated that it would be important to find ways to integrate the intervention into their current discharge procedures without adding to their work. However, physicians noted that the dearth of physicians in rural communities puts them under pressure to focus their efforts on diagnosing and treating patients, affording them little time to provide education about warning signs to patients. Moreover, they revealed that delivering the time-consuming education required by the intervention is at odds with their fee structure, which is based on the number of patients they process, not how long they spend with patients.

Participants noted that HCPs would need to provide patients with detailed information about warning signs, both before and after discharge, but were concerned that this would be too time-consuming. Consequently, participants recommended increasing HCPs' ability to provide the education by re-directing some of their non-clinical tasks to administrative assistants. Alternatively, participants suggested that the education could be assigned to a nurse with dedicated time to deliver the intervention. They added that clearly defined professional roles would enable them to deliver the intervention by delineating who would assume responsibility for which aspects of the intervention, within and across healthcare sectors (e.g., hospital, primary care, and home care).

**HCP communication.** Participants identified poor intra-sectoral communication (e.g., within the hospital setting) as a potential barrier to providing the intervention. Physicians noted that they do not communicate the discharge diagnosis with much lead time and that they would need to inform nurses which warning signs education to provide, which would add to physicians' workload. Physicians recommended creating a checklist with the most common medical conditions to streamline physician orders; they could check off the applicable medical conditions so nurses would know which

**Table 2. Healthcare Professional Demographic Characteristics.**

| Characteristic | N | % |
|---|---|---|
| **Gender** | | |
| Woman | 35 | 83.3% |
| Man | 7 | 16.7% |
| **Ethnicity** | | |
| Caucasian | 40 | 95.2% |
| Arab | 1 | 2.4% |
| South Asian | 1 | 2.4% |
| **Highest Level of Education** | | |
| Baccalaureate Degree | 16 | 38.1% |
| Master's Degree | 15 | 35.7% |
| College Diploma | 5 | 11.9% |
| Medical Degree | 5 | 11.9% |
| Other CCFP (CAC-COE) | 1 | 2.4% |
| **Professional Designation** | | |
| Registered Nurse | 11 | 26.2% |
| Physician | 7 | 16.7% |
| Social Worker | 6 | 14.3% |
| Nurse Practitioner | 5 | 11.9% |
| Occupational Therapist | 3 | 7.1% |
| Physical Therapist | 3 | 7.1% |
| Registered Practical Nurse | 3 | 7.1% |
| Registered Dietician | 2 | 4.8% |
| Rehab Professional | 1 | 2.4% |
| Rehabilitation Assistant | 1 | 2.4% |
| **Primary Position** | | |
| Staff Nurse | 8 | 19.0% |
| Nurse Practitioner | 6 | 14.3% |
| Physician | 6 | 14.3% |
| Care Coordinator | 3 | 7.1% |
| Occupational Therapist | 3 | 7.1% |
| Rapid Response Nurse | 3 | 7.1% |
| Social Worker | 3 | 7.1% |
| Physical Therapist | 2 | 4.8% |
| Registered Dietitian | 2 | 4.8% |
| Independent Contractor | 1 | 2.4% |
| Medical Resident | 1 | 2.4% |
| Nurse Educator | 1 | 2.4% |
| Patient Navigator | 1 | 2.4% |
| Rehabilitation Assistant | 1 | 2.4% |
| Team Leader | 1 | 2.4% |
| **Healthcare Sector** | | |
| Community | 27 | 64.3% |
| Hospital | 15 | 35.7% |
| **Region** | | |
| Northeast | 31 | 73.8% |
| Southwest | 10 | 23.8% |

*(Continued)*

**Table 2.** (Continued)

| | | |
|---|---|---|
| Other | 1 | 2.4% |
| **Work Hours** | | |
| Full-time | 40 | 95.2% |
| Part-time | 2 | 4.8% |
| **Age** | | |
| Median | 38 | |
| Range | 24-68 | |
| **Years in Current Position** | | |
| Median | 7 | |
| Range | 1-33 | |
| **Years Worked in Current Employment Location** | | |
| Median | 6.5 | |
| Range | 1-29 | |
| **Years of Professional Practice** | | |
| Median | 13 | |
| Range | 2-38 | |

*Note:* CCFP = College of Family Physicians of Canada; (CAC-COE) = Certificates of Added Competence in Care of the Elderly. Totals may sum to more than 100% because participants could select multiple response options.

education to provide. Other HCPs also recommended a checklist delineating the intervention steps and their timing (e.g., five tasks to be completed within 24–48 hours of admission), and serve as a communication tool between HCPs about what education has been provided, and what still needs to be provided and by whom.

Participants also highlighted communication issues across sectors (e.g., hospital and community) as potential barriers to delivering the warning signs intervention; numerous participants noted that HCPs working in different sectors frequently fail to communicate about the services provided to patients. For example, participants explained that community HCPs are not privy to what education is provided in hospital, and thus do not know what further education is needed in the community. The lack of a centralized or shared electronic medical record (EMR) was seen as a significant barrier in this regard. Participants explained that different sectors chart on different EMRs that cannot be accessed universally; HCPs in hospital have no access to primary care notes in the community, and community clinic HCPs cannot access hospital HCP notes. Participants thus recommended a universally-accessible EMR or mechanism to inform community HCPs that patients received the warning signs intervention during hospitalization and will require follow-up education at home after discharge.

**HCP knowledge and skill.** Participants foresaw HCP knowledge deficits of what constitutes a warning sign for a particular health condition as likely to impede delivery of the intervention. Knowledge deficits were expected to be especially problematic when it comes to patients with multiple chronic conditions in terms of knowing which condition was worsening, and patients with complicated conditions that are unfamiliar to HCPs. Participants suggested that patients with complicated conditions be diverted to HCPs with specialized expertise (e.g., an oncology nurse to help develop a warning signs intervention for patients admitted with cancer). Physicians viewed themselves as less skilled in providing patient education than they viewed other members of the interprofessional team and suggested that this would be a barrier to delivering the intervention.

Identified facilitators included training opportunities to enhance HCP knowledge and skill in delivering the intervention such as refreshers on the different medical conditions HCPs may encounter, and how to use the teach-back method. Participants suggested learning strategies such as mentorship and education should be offered to HCPs, particularly allied HCPs who were likely to require more foundational training about warning signs.

**Table 3. Exemplar Quotes.**

| Category/Sub-category of Barrier/Facilitator | Illustrative Quote |
|---|---|
| **1. Material Resources** | |
| Lack of educational materials would prevent HCPs from providing the intervention | AH4: I think a lack of resources is an overall problem, yes. Physical resources, yes, for sure. You know, it's one thing to sit there and to, you know, verbally educate a patient about the warning signs of their specific condition but, I mean, dependent on their cognitive level and other factors, they might not retain that, like I said. So, you know a written educational material for a specific condition would probably be ideal…I would say that's probably a problem. |
| Insufficient condition-specific teaching materials, unsure where to locate Only permitted to use employer-approved teaching materials, yet these were very limited | AH13: What I'm reading here, like, obtain and prepare written educational information on warning signs relevant to the patient's condition. And like, I would have to create that myself. Like I have no idea where I would access that information. AH13: I know that we get told that we're not really supposed to provide clients with any educational materials that are not [employer]-approved documents with our letterhead on it…it would be helpful if we actually had [employer]-approved documents that we were providing. |
| Need concise, pre-packaged written teaching materials for the medical conditions they most frequently encounter | AH4: Maybe just like a handout about, you know, you're, you've been discharged from the hospital with X, Y, Z, your, you know, things to look out for, making sure of this. Something that's very user-friendly that the patient can understand. Warning signs that would indicate you need to go back to see a doctor, things like that. |
| Materials should be available in different languages | N19: And having it in different languages, I mean it's simplified information but in the main languages that maybe suit your community, whether it be Italian, Portuguese. |
| Materials should contain colour-coded decision supports that designate urgency | N9: So green is, "This is how you should feel every day, this is what you should be doing." Yellow is "caution, if you start feeling like this, these are your action plans." Whether it be CHF" – I mean you're doing your weights and this and that every single day. But yellow is, "You had an increase in weight. This is your action plan." And then red is, "These symptoms, call 911, go to the hospital." And such fantastic tools, but again when I work in the hospital nobody's ever even heard of them. But if these action plans and identification of symptoms can be started in the hospital with a universal tool that everybody across the board is aware of, I think that's – again, one of the issues is that nobody overlaps. |
| Written materials could be supplemented with instructional videos that patients could begin viewing while in hospital | N30: Or even potentially while they're still in the hospital, maybe something on, like, a memory stick, right, where they can watch it…most hospital beds now have, you know, a TV or some kind of video device where you can easily just throw in a USB stick and they can watch a video, right? Because a lot of times, it's a lot of sitting and waiting in the hospital room until, you know, you're ready to be discharged. So, I mean, helping to A) pass the time that way, and B) kind of learning what to expect once you're discharged home, that might help, as well. |
| **2. Human Resources** | |
| Insufficient staff/time to implement the warning signs intervention | N12: Barriers….So in my current context would be just manpower issues. |
| Dearth of physicians in rural communities makes them focus diagnosing and treating | P1: we're always overworked and you know, we're overstretched, and this is going to add to our workload…we feel it's our responsibility to see as many patients as possible. So, given the workload that some people have, sometimes they feel like they should see, you know, the time is more efficiently spent seeing more patients…I think this goes back to my previous point, you know, maximally distributing energy and effort, and time. So, I think that's probably the biggest one. It's not that we don't like to teach, it's just there's not much time to do it… sometimes patient education has not been seen as the most prominent aspect of our job. So, diagnosing and managing - physicians have seen it [as the most prominent aspect of their job]. |
| Intervention at odds with physician fee structure | P1: And then you know the OHIP [Ontario Health Insurance Plan] fee schedule doesn't – you know all of that paperwork and all that back and forth you know in general doesn't really get paid very well. I think there's a one-time CCC [Complex Continuing Care] code and another code for communication but I may be off with that. But I'll just say, you know physicians are already strapped and a lot of the us don't – you know just to get paid for some of this may be more onerous kind of follow-ups to give that extra direction versus having them just come in, you know, potentially to the office or something… Because in Ontario, we get paid per person, not per time you know spent with patients in general. |
| Providing patients with detailed education may be too time consuming | N1: [Educating patients and families] it's really time-consuming. |
| Increase HCPs' ability to provide the intervention by re-directing some of their non-clinical tasks to administrative assistants | N12: They [leaders and administrators] need to be able to give you a strong secretarial sort of, whatever you're going to call it, medical secretary, administrator, who can, help you with med lists, calling patients, following up on like the different sheets that you're looking for, the different forms, all that stuff. |

*(Continued)*

| Category/Sub-category of Barrier/Facilitator | Illustrative Quote |
|---|---|
| Increase HCPs' ability to provide the intervention by decreasing bureaucracy | P3: In terms of our load, I mean, honestly, I spend far more time looking after sick computers than patients, and they keep telling us that they make our lives easier, and I am not impressed. [Laughs] And I don't know what to do about that, but it just seems like, well, just do it on the computer, and…it's just, like, more and more is getting downloaded that way. And oh, you need to fill another form out to make this happen, and oh, you need to sign three more forms from here to make that happen, and now it's a CT scan to order it, you don't just need one form, you need two to make sure you're not ordering too many. And it's just becoming ridiculous. And some forms go through the computer and some don't, like, I don't know what – just, does Administration recognize the bureaucracy that's going on of effective clinicians in paper and computer ridiculousness? |
| Education could be assigned to a nurse with dedicated time to deliver the intervention | AH3: downloading this on to the nurses is going to be really difficult whereas if you have…a sole case manager whose role is to say "I'm going to help you get through this, from here to home and when you're home we'll make sure things are running smooth." So, to have a solo case manager-type role, it really makes it easier for that person to be more effective than someone under the crunch of nursing. |
| Clearly defined professional roles would facilitate HCPs' ability to deliver the intervention | AH1: If we let physio do their job, social work do their job, OT do their job and the case manager fills in…then AH2 can get her part, AH3 can do his and I as a case manager can just do the generalized education, make sure that they're looking for the warning signs in addressing the exercise that AH2 does and looking at the mental health components that AH1 might have identified and it just works better because…I'm allowing physio to do their part because they don't have to worry about this component. |
| **3. HCP Communication** | |
| Intra-sectoral communication | P1: You know we have to tell the nurse what they need to teach…And so is that going to be an easy thing to do?…Having people who are going to do these sessions - who is going to feed them [nurses] the information? Are they [nurses] going to try and interpret what a physician's discharge note is? |
| Discharge diagnosis not communicated with much lead time | P3: I've discharged the patient…I've dictated a discharge summary. So the nurse has the meds, but the nurse may not even have all the discharge diagnoses, because I've dictated the discharge summary, I haven't written it. |
| Implement an order set checklist or checklist with intervention steps and timing | N30: Like a checklist or processes or something where it's, like, OK, you know, within, you know, 24 hours of admission, these are, like, the five steps or the five tasks that need to be done, and then during the stay, again, having a checklist. Like, so, any person – ideally, if it's the same provider, great, but, I mean, if another provider were to pick that up, knowing exactly where along kind of the care plan that patient is at. |
| Inter-sectoral communication | AH13: the continuity of information that's provided, and I think that's the missing piece is that we're not all [hospital and home care] using the same resources [communication systems]. |
| Lack of a centralized electronic medical record | AH13: Communication is a large barrier to implementing this. Especially because there is quite a lack of access between hospitals and community care in terms of information on what actually happened in the hospital. Like our IT [information technology] systems are totally different and not linked. So sometimes we receive, like, no information at all in community care about what occurred in the hospital other than word of mouth from the client. |
| Need a mechanism to enhance communication | N5: So it [a mechanism for communication] might help with communication saying yes, this patient did view these videos regarding their condition so the nurse in home would already have an understanding of what's being reviewed in hospital. |
| **4. HCP Knowledge & Skill** | |
| Knowledge deficits of what constitutes a warning sign for a particular health condition. | AH4: I might not be able to be fully knowledgeable on educating people of warning signs related to a specific health problem. |
| Knowledge deficits surrounding which chronic condition is worsening | P3: people aren't exactly sure even what they were in the hospital with. And frankly, in complex geriatric patients, sometimes I don't know what they were in the hospital with, right, because they had a cascade of events, you know. |
| Conditions with which HCPs are unfamiliar | N13: But it would be the very random kind of sporadic things like, "Oh, I've never seen this [condition] before." We're always learning in nursing so you'd maybe have to teach yourself a little bit before you taught anybody else. |
| Physicians viewed themselves as less skilled in providing patient education | P1: When patient education is so hands-on and one-on-one…often physicians will say, you know, "I'm not the best equipped at doing this…leave it to the patient's nurse who probably knows the patient best." |

*(Continued)*

**Table 3.** (Continued)

| Category/Sub-category of Barrier/Facilitator | Illustrative Quote |
|---|---|
| Education for HCPs needed on the different health conditions they may encounter | P2: They [the HCP] has to learn the warning signs of impending complications. What are the signs? OK, so she just has to know those signs. And when we talk about what are the warning signs, when obviously the warning signs will be different case to case, or I should say from diagnosis to – with different diagnoses. So the nurse will have to learn the warning signs of complications in the case of the patient who has just had a heart attack and he's leaving the hospital. What are the warning signs of a complication due to a heart condition? But the patient who has chronic airway disease and the lungs are not good, then she would have to learn another set of warning signs for the complications of the lung issues. The patient who was in hospital because of the kidney failure, or the patient who was in hospital because of a diabetes flare-up of some kind, these, too, are yet again another bunch of warning signs that they have to look for when they have been discharged. They [the HCP] is going to learn the warning signs of impending complications, but those warning signs are different from one condition to other conditions…They have to understand those warning signs for every condition. |
| Mentorship and refreshers | N24: Reviewing with staff…because I think everybody can always use a kind of refresher…providing healthcare workers with, offering them access to…a webinar or in-service or something like that, I think that would just help.<br><br>N1: Mentorship could help…. |
| How to use teach-back and particular teaching materials | N1: I think having mandatory education for these people [less experienced HCPs] to demonstrate that they have the ability to identify and have good teach-back and they're teaching the appropriate skills to help manage their disease processes. |
| **5. HCP Buy-in** | |
| Buy-in a challenge with any new initiative | N21: Yeah, I think that's [buy-in] always a factor with everything. You know, I don't think – I think it's difficult to get buy-in across the board with anything in health care. And, yeah, like, I think, you know, I find especially – and I can, like, speak to, like, where I work, but, like, where I work, you know, for example, the physicians, like, things really need to be clear and clearly laid out and very concise before they're confident in, you know, that working and, say, referring patients to those things. |
| HCPs unwilling to invest time in initiatives that may not be sustained | P3: Country docs with very few resources, who historically have tried to make some of this happen and it hasn't. So there's a lack of trust in a new way. |
| Burnout from the pandemic era may also undercut HCP buy-in | N24: Another thing is burnout of nurses during Covid, too. Like, to add on another task or to add on something to their workday. |
| Buy-in from nurses | N10: Nurses, if they think something is not something they want to do, then it's going to be a hard thing for them to do. But like I said earlier, for something like education we would love to do it. I think most nurses really like that piece or at least see the importance of it, but getting them to buy in to the fact that they have time and this is going to be a fairly easy and smooth transition to change their practice to do this, that would probably be the hardest part. Because it might be hard for them to see that is all very doable if you had it laid out nicely that it was doable. But just keeping everything the same and saying, "Okay, now we need you to do all this teaching," that would be a hard one for nurses to follow I think, and accept it. |
| Buy-in from physicians | P1: My biggest thing is getting the buy-in of the physicians, and say it's efficient. Because I'm on tons of these groups and policies and they [policies] come from up top, from admin [administration]. And they [administration] you know think it's all great ideas, and then they forget to ask the people in the trenches of like "is this good? How would this work? What would you change to make it more efficient for you in the trenches to be efficient in your day?" |
| Using evidence to convince stakeholders to buy in | AH16: I think people want to see that it works [the empirical evidence] before they buy into it. |
| Identifying champions | P3: Yeah, well I think the first thing is you need to have – identify those champions, those people that are comfortable doing it and understand it. |
| **6. Context of Rural Practice** | |
| Long distances and inclement weather regularly prevent HCPs from conducting home visits | AH2: Most of my patients…definitely live outside that 30- minute radius, and then again, they're – so they're in an isolated community, maybe with very, very limited social networks and relying on friends and neighbors maybe giving them a ride home from the hospital and stopped at the pharmacy to pick up their stuff, dropped them off and said "OK bye." They're the ones who don't qualify for the rapid response in-home nurse to see them…the very rural people…I think for me it's the – like the actual physical isolation and the remoteness of some people after they're discharged from hospital. |

*(Continued)*

**Table 3.** (Continued)

| Category/Sub-category of Barrier/Facilitator | Illustrative Quote |
|---|---|
| Difficult for physicians to predict in which setting they will be working when a patient is discharged | P3: The reality of my life is that I've got my hospital in-patients… I've got my family medicine patients and all various stages of mental health issues that may call me to emerge or what have you, right. Like every week is a different gong show…You can't say, oh, Mister So-and-So, I'm sending you home with your congestive heart failure that has resolved, and I will be checking in on you in two days' time. Because I really don't know what two days from now is going to look like in terms of my availability to get out there or what have you. |
| Physicians felt practicing in varied settings would foster their ability to provide the intervention | P2: There's an advantage in our setting [rural] where I work, that I'd probably be taking care of the patient both in the hospital and communicating with them at home, and so that would be an extra point of contact for them if they needed some extra help in managing things at home. |
| The small size of rural communities makes HCP familiar with patients | P2: People in rural areas, there is a small number of patients and a small number of providers. So, in the end, people are familiar with each other. People are familiar with each other from the grocery store, from walking in the street. Like, it's a small number of people. So, when they get sick they see the nurses, and when they get vaccinated they see the same nurses each, somewhat. So, they know each other and they're familiar with each other. |
| **7. Patient- and Family-Specific Characteristics** | |
| Low health literacy | P2: What can inhibit this intervention? The level of literacy of the patient or the family – that sometimes is very important also. Here I'm going to say that the fact that she [nurse] has to tell – to address the family and the patient in a way proportional to their level of literacy is very important. Why? Because when they – they say a little knowledge is a bad thing, meaning if I tell you…"Is it serious?" They ask, "Is it serious?" And if you say well, it can lead to gangrene, it can lead to an amputation of your leg, it can lead to abscess and poisoning in your whole system. If the patient is not able to differentiate between [those things], all they will retain in their mind is "I'm going to lose my leg. I'm going to get poisoning, and there's no hope, and I might as well die now," as opposed to absorb the fact that she [the nurse] is simply listing different complications that could happen – not that will happen. So I guess what could stop this [the warning signs intervention] from happening and being useful is if the level of literacy of the patient and family is not very good. Then maybe you will need someone else again from more distant family or something, to intervene and try to explain or in a way try to translate what she's trying to say. That's one thing we find. That a lot of people you tell them things and then when you see them again, they repeat what they understood. And it's totally different from what you told them. |
| Hearing or language barriers | AH2: People with difficulty hearing, it's really difficult. Or even language barrier, that kind of thing, if English isn't their first language. |
| Cognitive impairment | N24: If there's some cognitive issues…and not retaining the information I find I'll see them in the clinics, for example, following a discharge and they don't even know where their discharge papers are or where the instructions are. And it's like they need prompting them to be like "ok, well what were you told? Were you supposed to go and check-up, what were your follow-ups?" And these people…don't really know what the next steps are. |
| Patients and families overwhelmed with hospitalization | AH4: I think sometimes it might be overwhelming though because they're in hospital and, you know, it could be overwhelming and they may forget, you know, the retention might not be there. That might be one negative depending on how much more, you know, how much more stuff is involved like doctors going in and out, nurses, there's other people. So it would be more just potential for, just the retention… they're dealing with the actual acuteness of the health condition itself as opposed to necessary thinking about discharge, you know. |
| Complexity of patients' conditions | P3: people aren't exactly sure even what they were in the hospital with. And frankly, in complex geriatric patients, sometimes I don't know what they were in the hospital with, right, because they had a cascade of events, you know. |

*(Continued)*

**Table 3.** (Continued)

| Category/Sub-category of Barrier/Facilitator | Illustrative Quote |
|---|---|
| 8. Risks & Liabilities | P1: I think multiple patients…don't have, is understanding sort of the indications of coming back into hospital, what part of their medical – what part of the issue is actually related to their condition, and what part of it actually can be dealt with at home, and what are their responsibilities, in fact, of maintaining and avoiding coming back to into hospital.<br><br>I think part of the problem is that we're all [physicians] also medically and legally inclined to just say, "Go back to emerg," and I remember that being also, sticking in my head. The model itself is great, but that it will require a bit of courage…Take someone with COPD for instance. COPD is a chronic medical condition, often has a baseline cough, and these patients will often be short of breath, depending on the severity of the disease. And they don't necessarily need to come to hospital if they take their puffers correctly, but then we always tell them when they go home that if the cough gets worse, or you know, there's a fever, that they should, you know – or if they feel more short of breath then they should come to hospital. I feel that's sort of a legal way to cover your practice, but at the same time, it's not particularly helpful with such a vague piece of advice to patients. I felt that physicians always have that idea in their head, well, at least from my perspective, that they want to be covered medically and legally to say, "You know what, worst case scenario, come back to emerge," as opposed to, you know, "Just increase your puffers or do this or do that." |
| 9. Timing of Intervention Delivery | P2: It [logic model] says that she [nurse] has to get involved within the first 24 hours. What warning signs are already known by the patient and family? What do they know and what needs to be learned? Now, yes, I have a reservation. Believe it or not, when a patient gets admitted, in the first 24 hours no one knows what's wrong with the patient…When a patient gets admitted, it's usually he has chest pain, some X-ray, some blood work has been done, some cardiograms have been done. "It sounds like it's his heart. Let's admit him. Let's stabilize him." So he gets admitted. No one knows yet: is this a heart attack, or is this a blood clot, or is this an anxiety episode? They don't know. No one knows. So this girl [nurse] – I don't think it's useful for her to go in the first 24 hours – maybe not before the third day. I'll tell you why. Because six o'clock in the evening, that's considered day one. They admit him to stabilize him, and they put him in a bed, and they hook him up to a few things, and then day two in the morning comes a physician who will look after him in the hospital. He starts to look at some results that have been done of tests, and orders a few other tests, and speaks to a couple of specialists on the phone to get their opinion as to what they think should be done. And then by the third day, you start to get more feedback from the tests and a diagnosis starts to become likely or obvious that we're dealing with such-and-such. So with the intervention, don't get her involved in the first 24 hours after admission. Get her involved in the third day. By the third day it's become obvious what – it has become obvious what's going on. So at that point she can talk to the patient and the family and tell them it seems that the diagnosis is such-and-such. What do you know about his condition? Here are some warning signs of complications that may happen in the future on discharge. |

*Note:* HCP = healthcare professional.

**HCP buy-in.** Nurses and physicians noted that, as with any new initiative in health care, getting buy-in from all potentially interested parties (e.g., clinicians, managers, administrators) is likely to be a barrier. One physician discussed how some HCPs are skeptical of new interventions because they had collaborated on prior ones enthusiastically, only to feel like their efforts were wasted when the interventions were discontinued. Another physician and other nurse participants noted that burnout from the pandemic era may also undercut HCP buy-in. However, nurses expressed that achieving nurse buy-in, specifically, should not be difficult because they would recognize the potential benefits of the warning signs intervention but with the caveat that their non-clinical tasks would need to decrease to allow them to incorporate it into their practice. Participants spoke about strategies to cultivate buy-in such as using evidence, prior to implementation, to convince all interested parties that the intervention will benefit both patients and HCPs. They noted the importance of fostering long-term buy-in by identifying champions and trialing the intervention with them to iron out kinks prior to more widespread implementation.

**Context of rural practice.** Participants outlined numerous potential barriers to implementing the warning signs intervention in the context of rural practice. For instance, nurses and allied HCPs expressed that although home visits are preferable to the telephone when providing education (particularly for patients with hearing difficulties or cognitive impairment), long distances and inclement weather regularly prevent HCPs from conducting home visits. Moreover, nurses expressed that patients who live far from the hospital may be ineligible for a home visit because some programs, such as the rapid response nursing program that currently exists to address the needs of high-risk patients following hospital discharge, do not allow home visits to patients beyond a 30-minute drive from the hospital.

Some physicians explained that they practice in hospitals, clinics, and patients' homes, which makes it difficult to predict in which setting they would be working when a patient is discharged or if they can conduct a home visit. Another physician noted that practicing in these varied settings would foster their ability to provide the intervention because they would benefit from knowing what had been provided in hospital when the patient returned home.

Participants had little to say about facilitators to the intervention in the rural context, but did observe that the small size of rural communities means that HCPs are often familiar with patients, both in a health and a social context (e.g., they see them in the grocery store), and that this rapport could facilitate HCPs' ability to deliver the intervention.

**Patient- and family-specific characteristics.** Participants underscored that low patient and family health literacy might interfere with uptake of the intervention and suggested that finding ways to simplify warning signs education may be crucial. Participants underlined how cognitive impairments, hearing difficulties, and language barriers that some patients and families experience can interfere with HCPs' ability to deliver the intervention, and how severity of illness or being overwhelmed by a hospitalization may interfere with patients' and families' ability to grasp and retain information about warning signs. Physicians noted that the complexity of patients' conditions may make it difficult to discern precisely which condition was responsible for the hospital admission and, consequently, what condition-specific warning signs education is required. In terms of facilitators, participants emphasized that families who are involved in a patient's care can help the patient and other family members understand what warning signs to monitor.

## Barriers and facilitators identified by one or two HCP strata

**Risks and liabilities.** Participants in the physician and allied HCP strata identified that the risk of legal implications could prevent them from implementing the intervention. Both strata expressed reticence about permitting patients and families to audio- or video-record the education provided on the warning signs. For example, one physician expressed concern that physicians could be at risk of being sued by patients or families who misunderstand the warning signs education and do not follow through with the recommended medical advice. This physician suggested that this risk could possibly be mitigated by establishing a system that enables patients to reach out for guidance after discharge if they are unsure about what to do.

Several allied HCPs expressed concern that they could risk disciplinary action from their professional regulatory College if they were to teach patients about what warning signs to monitor and what to do about those that are detected because this was not within their scope of practice. They proposed that healthcare administrators should not expect allied HCPs to deliver the warning signs intervention but rather to support the education being delivered by nurses or physicians.

**Timing of intervention delivery.** The timing of initiating the warning signs intervention was seen as a likely barrier to its implementation in the hospital. A physician noted that initiating the intervention within 24 hours of admission would be unrealistic because, in some cases, the full diagnosis will not yet have been established; therefore, assessing patient and family health literacy should likely wait until the diagnosis is confirmed. Nurses working in community clinics expected to have difficulty following up within 24–48 hours of discharge because their patient rosters would already be full. Participants made few suggestions in terms of facilitators, though it was recommended that nurses could introduce themselves to patients within 24 hours of admission to establish rapport before assessing health literacy pertaining to relevant warning signs.

## Discussion

Our findings shed much-needed light on barriers and facilitators that should be considered to ensure successful implementation of the warning signs intervention in rural TC. Such understanding is needed for successful implementation [40]. Below, we discuss our key findings in the context of the existing literature and their implications for future practice, policy, and research.

In terms of human resources, it is notable that early discharges and shorter hospital stays are a worldwide trend. Since 2011, the average length of hospital stay has decreased in 36 of the 38 Organization for Economic Cooperation and Development countries [16]. Current emphasis on early hospital discharge requires HCPs to deal with high patient turnover, which increases workload related to admission and discharge [41]. Our sample of HCPs had an average of 14 years working in their professions. Consequently, they have witnessed firsthand the trend toward shorter hospital stays – and the associated increased patient turnover – and are familiar with the challenges of devoting more of their workday to preparing patients for discharge and/or following up with them after discharge. It is therefore not surprising that HCPs identified that human resources would be a barrier to delivering the warning signs intervention and suggested ways to enhance their ability to provide the intervention, such as reallocating their non-clinical tasks to administrative personnel.

Other research has found that costs remain a barrier to hiring the additional staff required to deliver TC interventions [40,42]. Nonetheless, evidence suggests that the increased cost of implementing the warning signs intervention may be countered by cost recovery from consequent reductions in hospital readmissions [43].

The issue of financial constraints was also evident amongst physicians who identified that the time to provide the education required of the warning signs intervention is at odds with their remuneration structure as well as expectations of quickly diagnosing and treating patients. These findings concur with other research emphasizing that remuneration is an obstacle to physicians adopting new practices. Mitra et al., for instance, emphasized that Ontario physicians view fee-for-service models as incentivizing them to quickly treat high volumes of patients and were frustrated that fee-for-service structures did not keep pace with practice change expectations [44]. Our study demonstrates that physicians are not incentivized to provide the intervention, but also underscore an ethical tension between physician financial remuneration policies and best practices. It is worth pondering what harm might come to patients when physicians do not provide the warning signs intervention, and the potential costs in the form of readmissions, adverse events, as well as burnout of physicians who may feel they are not providing the best possible care.

Our findings on HCP knowledge and skill highlight the training that HCPs require (e.g., what constitutes a warning sign for a particular health condition, how to use the teach-back method) and that allied HCPs, in particular, may benefit from training to successfully implement the intervention. Other research has likewise reported that failing to provide initial

training in implementing new TC interventions impedes implementation [45]. Also, TC interventions in research studies are typically delivered by HCPs with additional training [10]. Consequently, providing HCPs with additional training should be a standard part of implementation; failing to provide an intervention as originally designed can result in low intervention uptake and effectiveness [46]. However, we found little research identifying what knowledge and skills HCPs need to successfully implement warning signs interventions.

A novel finding of our study is the need for HCP training on what constitutes a warning sign for a specific condition and the need for standardized, condition-specific (e.g., Congestive Heart Failure, Chronic Obstructive Pulmonary Disease) warning signs teaching materials. While such materials were proposed for patients and their caregivers, these could also benefit HCPs who lack knowledge of the warning signs of a particular health condition. Other research has likewise reported that HCPs' ability to implement TC interventions is facilitated by their access to related knowledge and information [18].

Previous research has underscored that HCPs are often unable to prioritize TC because of high patient acuity and limited time, which challenge communication [47]. For example, hospital-based HCPs may experience difficulties coordinating care amongst interprofessional team members who work in the community [47]. TC requires different team members to communicate, which may be challenging in fast-paced practice environments [48]. Consequently, it is understandable that intra- and inter-sectoral communication was seen as a potential barrier to providing the intervention. Our findings illuminate potential facilitators that may mitigate this barrier, such as using checklists as communication tools among HCPs to track what warning signs education still needs to be provided and who is responsible for doing so. Although checklists and communication tools have been identified as facilitating TC more generally [17,18], to the best of our knowledge, they have not been identified as facilitating intra- and inter-sectoral communication surrounding warning signs interventions.

In the same vein, participants' discussion of the lack of a shared electronic medical record (EMR) as a significant barrier to intersectoral communication stands to further complicate the sharing of information. In Canada, health care is administered by its 13 provinces and territories [49]. In the province of Ontario, not only is there variability among the EMRs that are used across sectors, but there is also variability among EMRs within a given sector because healthcare organizations are generally free to choose which EMR they use based on minimum technical specifications and provincial pre-certification. For example, OntarioMD lists 13 different EMR versions or platforms as certified for use in primary care as of its last posted update on March 26, 2024 [50]. We are not aware of any current policy direction in Ontario to consolidate EMRs or mandate a single EMR for use across all sectors. Therefore, our finding that different sectors chart on different EMRs and thus cannot share notes with each other adds to the evidence suggesting that a major policy shift is needed that prioritizes inter-sectoral communication. This is especially important given that research conducted in Denmark has identified that shared EMRs improve intersectoral HCPs' ability to communicate in transferring information between sectors [51]. Other research has identified that EMRs are underutilized but, unlike our study, did not pinpoint the reasons why [10].

Regarding risks and liabilities, a unique finding of this study is that allied HCPs were concerned about disciplinary action from their professional regulatory College if they delivered the warning signs education because they perceived it as beyond their scope of practice. They maintained that healthcare administrators should not expect allied HCPs to deliver the warning signs intervention independently, but rather have allied HCPs support the education delivered by nurses or physicians. It is plausible that allied HCPs were resistant to performing tasks that, based on their training and professional socialization, they see as beyond their scope of practice. Prior research has identified that it is stressful for HCPs to be redeployed to new units or areas that are outside their scope of practice or beyond their usual duties [52]. Yet, previous research, such as a systematic review conducted by Leong et al. (2021) has also stressed that allied HCPs can successfully learn to adopt expanded roles for tasks such as patient and family education [53].

Both allied HCPs and physicians also expressed liability concerns about permitting patients and families to audio- or video-record the education provided about warning signs. However, liability insurers construe recordings as a safeguard

to protect physicians from malpractice lawsuits [54], and there are few legal grounds to refuse patient requests to record their conversations with medical professionals [55]. In contrast to physicians and allied HCPs, nurses did not raise concerns about risks and liabilities. Although further research is needed, it is plausible that, unlike allied HCPs, nurses saw the warning signs intervention as within their scope of practice, which may explain why nurses did not identify risks and liabilities as a barrier.

Our findings that achieving buy-in from all interested parties (e.g., clinicians, managers, administrators) may be a barrier and that providing evidence on the potential outcomes and targeting champions to trial the intervention may facilitate widespread uptake generally concurs with other research [40,45]. However, a novel finding of our study is that nurses expressed that achieving nurse buy-in should not be problematic because they would recognize the potential benefits of the warning signs intervention. This finding suggests that potential champions to lead implementation of the warning signs intervention can be found amongst nurses.

Finally, our finding that aspects of the rural context – specifically, long distances and inclement weather preventing home visits – present barriers to delivering the intervention both reinforces and expands our understanding of rural contexts of practice. On the one hand, this finding highlights the persistence of longstanding challenges tied to rural geography and insufficient rural human health resources, which are well-documented in the literature [56–58]. On the other hand, our study shows how the rural context extends to hospital-to-home TC specifically, which is only beginning to be examined within the literature [12,26,59–61]. It is important that rural patients know what warning signs to monitor because between 23% and 33% of patients experience complications within six weeks of hospital discharge [62].

## Strengths and limitations

Our study addresses a significant gap in TC research. A recent scoping review identified that 11 studies have explored barriers and/or facilitators to implementing hospital-to-home TC; however, all focused on TC in general [45]. None examined the warning signs intervention component of TC or were conducted in rural settings, and very few included physicians [45]. Since most prior TC research has not included physicians [45], we believe their inclusion, although limited, was a major strength of our study. Our study also included HCPs from different professions and sectors, which provided an enhanced, interprofessional understanding of barriers and facilitators to implementing the warning signs intervention. It is one of the few studies that focus exclusively on rural communities contributing to a small but growing interest in rural healthcare professionals' perspectives on barriers and facilitators to implementing interventions.

In terms of limitations, although the allied HCP strata was comprised of a variety of professionals (e.g., social workers, occupational and physical therapists), we did not directly compare their perspectives. It is possible that apprehensions about lacking knowledge and skill to deliver the intervention may differ across professions. For example, this concern may be more common for social workers than physiotherapists. The same can be said for worries about risking disciplinary action from professional regulatory Colleges. The study was conducted in healthcare jurisdictions in one Canadian province; however, the rural and rural-serving jurisdictions from which we recruited are typical of rural health care services in rural North America [63]. Consequently, the findings are likely to be highly transferrable to other rural jurisdictions facing similar healthcare challenges.

## Implications for practice

This paper constitutes a first step in the knowledge mobilization process for the warning signs intervention. The next step is to develop an implementation strategy, beginning with the adaptation of the intervention to fit the needs and resources available in rural contexts. While we are conscious that some of the categories of barriers and facilitators are more actionable than others, we suggest that categories identified by all three HCP strata be prioritized for intervention adaptation before incorporating the warning signs intervention into rural TC. The implementation plan should be guided by an evidence-based, ecological framework that emphasizes the importance of local context [64]. Regarding resources, the findings suggest that HCPs could benefit from standardized educational warning signs materials for different health conditions that account for hospital

readmissions. Some teaching materials are available (e.g., Congestive Heart Failure) [65] and these need to be identified, evaluated for their evidence, and mobilized into practice. Where teaching materials are unavailable, these need to be developed. HCPs could also seek out training on the warning signs of conditions that they most frequently confront in their practice.

Regarding context of rural practice, telephone follow-up may be a viable alternative to home or out-patient follow-up, as other research has identified that telehealth is feasible in rural health care [66]. Telephone follow-up has also been found to be effective in reducing hospital readmissions [9,67] and has been successfully implemented by hospital nurses serving rural communities [60], highlighting its potential for wider scalability.

Given that rural dwellers expect to be prepared to detect and respond to signs of worsening health conditions because of their limited access to HCPs and the distance they are required to travel to hospitals [12,13], attention to addressing barriers to implementing the warning signs intervention in practice is urgently required.

## Implications for policy

One impetus for this study was healthcare administrators' acknowledgement that HCPs, other than nurses and physicians, may be expected to deliver the warning signs intervention due to the dearth of human health resources in rural communities. Although we recognize that all allied HCPs may not be able to deliver the warning signs intervention independently, it is important to understand that the intervention is interprofessional and hence, all team members can contribute to its implementation [68]. Allied HCPs could support the implementation of the intervention in numerous ways, such as reviewing with patients the education provided by nurses or physicians. However, it is imperative that administrators ensure adequate staffing if they plan to implement a new role for allied HCPs in providing the warning signs intervention to give allied HCPs sufficient time to become competent in this role [69].

Additionally, research indicates that HCPs can learn to deliver new interventions that were not part of their initial training [70]. Members of specialized geriatrics teams, for instance, have been taught to perform geriatric assessments that were not previously part of their job responsibilities [70]. Accordingly, administrators may create cross-professional training opportunities on the provisioning of the warning signs intervention. Such interprofessional approaches are particularly important for rural healthcare provision given the already limited human health resources [71].

Lastly, policy-makers can use our findings to reallocate HCPs' non-clinical tasks to administrative personnel and to prioritize inter-sectoral communication by linking community and hospital EMRs. Furthermore, such linked EMRs would allow not only for better inter-sectoral communication, but also for performance measurement to foster improvement in real time [72].

## Implications for future research

Studies with larger sample sizes of physicians are warranted, as are studies exploring ethical considerations related to physician compensation. Future research is also needed to explore the perspectives of the different professionals that comprise the allied HCP strata. Because studies on the effectiveness of warning signs interventions in rural communities have been limited, effectiveness studies are needed as are studies evaluating the effectiveness and feasibility of participants' suggestions to improve their ability to deliver the intervention considered in this study. Research is needed on how allied HCPs can support the implementation of the warning signs intervention. Lastly, future research may explicitly sample HCPs from different types of rural communities (e.g., urban adjacent, rural agricultural, rural recreational/cottage county, remote rural) to validate our findings across the full range of rural health care service settings.

## Acknowledgments

We thank our HCP participants for sharing their perspectives with us. We also thank Dr. Alex Peel and our other knowledge-users for supporting the study and facilitating recruitment. We also thank Shannon Gordon and Igor Kabanov for their assistance in conducting the literature review for this paper.

## Author contributions

**Conceptualization:** Mary T. Fox, Jeffrey I. Butler, Sherry Dahlke.

**Data curation:** Jeffrey I. Butler.

**Formal analysis:** Mary T. Fox, Jeffrey I. Butler.

**Funding acquisition:** Mary T. Fox.

**Methodology:** Mary T. Fox, Jeffrey I. Butler.

**Project administration:** Mary T. Fox, Jeffrey I. Butler.

**Supervision:** Mary T. Fox.

**Validation:** Mary T. Fox, Jeffrey I. Butler.

**Visualization:** Mary T. Fox, Jeffrey I. Butler.

**Writing – original draft:** Mary T. Fox, Jeffrey I. Butler.

**Writing – review & editing:** Mary T. Fox, Jeffrey I. Butler, Adam M.B. Day, Evelyne Durocher, Sherry Dahlke, Mark W. Skinner, Janet Yamada, Ilo-Katryn Maimets.

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
