## [Decision Letter · Decision Letter 0]

17 Dec 2024

PONE-D-24-37531Healthcare professionals' perspectives on barriers and facilitators to implementing a warning signs intervention for older rural-dwelling medical patients at risk for hospital readmissionPLOS ONE

Dear Dr. Fox,

Thank you for submitting your manuscript to PLOS ONE. After careful consideration, we feel that it has merit but does not fully meet PLOS ONE’s publication criteria as it currently stands. Therefore, we invite you to submit a revised version of the manuscript that addresses the points raised during the review process.

**ACADEMIC EDITOR: **

I have carefully reviewed the manuscript titled 'Healthcare professionals' perspectives on barriers and facilitators to implementing a warning signs intervention for older rural-dwelling medical patients at risk for hospital readmission' submitted to PLOS ONE, and I find it suitable for consideration. However, addressing a few concerns could significantly enhance the overall quality and clarity of the study.

Major Concerns:

Ethics Statement:

The ethics approval is mentioned on page 5, and the statement indicates that written informed consent was obtained from all participants. It aligns well with PLOS ONE's requirement for transparency about ethical considerations. However, the statement could be more explicit about the nature of the informed consent process (i.e., whether it was written or verbal).

Data Availability:

The manuscript confirms that all data are fully available without restriction, as per PLOS ONE's data sharing policy. However, specific details on the repository or data accessibility may be needed. As stated on page 7, it confirms that "All relevant data are within the manuscript and its Supporting Information files." Ensure that no conflicting statements appear within the manuscript, such as stating that data is available on request elsewhere.

Methodology section:

The methodology on pages 13-17, detailing the qualitative descriptive design, is well laid out, but additional explanation might be needed regarding how the sample size of 42 participants was determined. This should be explained to ensure it is consistent with typical qualitative research expectations (heterogeneity of the respondents), as qualitative sample sizes are often determined by saturation rather than pre-set numbers.

Result section:

The presentation of results, including tables on pages 19-20, is thorough. However, additional clarifications might be necessary in terms of how the data was analyzed and how different strata (nurses, physicians, allied health professionals) were compared. Ensure that the methods section clearly explains the data coding and analysis processes, especially the use of NVivo 12 and qualitative content analysis, and the role of the two coders (pages 15-17).

Minor Concerns:

Abstract:

The abstract is clear, but it could benefit from a stronger emphasis on the specific outcomes or findings of the study rather than focusing predominantly on the background and aim. The key findings could be briefly summarized to help the reader understand the significance of the barriers and facilitators identified.

Results Table (Table 1 on page 19):

While the table is informative, a brief explanation in the text regarding the presence or absence of certain barriers and facilitators across different strata could enhance clarity. For example, there could be a deeper dive into why certain barriers were not identified by specific strata (e.g., "Risks & Liabilities" not being a concern for nurses).

Citations:

There are some references to literature in the introduction (pages 9-12), but more recent sources may be required to ensure the study is grounded in the latest research, particularly concerning rural healthcare and warning signs interventions.

Grammar and Formatting:

Some minor formatting inconsistencies are noticeable, such as the inconsistent use of "HCPs" (Healthcare Professionals) and the way quotes are integrated into the text. Ensure that all acronyms are defined on their first use and that the flow of written text remains coherent.

Conclusion section:

The manuscript could strengthen its conclusion by linking the barriers and facilitators to practical recommendations for policy implementation or future research, especially since the findings could have a direct impact on improving transitional care in rural settings.

We look forward to receiving your revised manuscript.

Kind regards,

Mohd Ismail Ibrahim, MCom.Med

Academic Editor

PLOS ONE

Journal Requirements:

 “This work is supported by the Canadian Institutes of Health Research, Funding Reference Number 163072, awarded to M.T.F. The Canadian Institutes of Health Research had no role in the design data collection and analysis, decision to publish, or preparation of the manuscript. https://www.cihr-irsc.gc.ca/e/193.html”        

3. We note that your Data Availability Statement is currently as follows: “All relevant data are within the manuscript and its Supporting Information files.”

Please confirm at this time whether or not your submission contains all raw data required to replicate the results of your study. Authors must share the “minimal data set” for their submission. PLOS defines the minimal data set to consist of the data required to replicate all study findings reported in the article, as well as related metadata and methods (https://journals.plos.org/plosone/s/data-availability#loc-minimal-data-set-definition ).

If your submission does not contain these data, please either upload them as Supporting Information files or deposit them to a stable, public repository and provide us with the relevant URLs, DOIs, or accession numbers. For a list of recommended repositories, please see https://journals.plos.org/plosone/s/recommended-repositories .

Reviewers' comments:

Reviewer's Responses to Questions

**Comments to the Author**

1. Is the manuscript technically sound, and do the data support the conclusions?

Reviewer #1: Partly

2. Has the statistical analysis been performed appropriately and rigorously? 

Reviewer #1: N/A

3. Have the authors made all data underlying the findings in their manuscript fully available?

Reviewer #1: No

4. Is the manuscript presented in an intelligible fashion and written in standard English?

Reviewer #1: Yes

5. Review Comments to the Author

Reviewer #1: The introduction highlights the unique challenges faced by the elderly as they transition from hospital to home care and thus provide valuable contribution to the literature by identifying the barriers and facilitators to implementation of health care interventions in this regard in rural settings. There is need to emphasize the uniqueness of the research and its applicability across other similar settings in order to fully understand the study's importance and extent of the research.

Qualitative methodology was an adequate scientific methodology for this study. However, how were the study participants selected (was it a purposive or convenience sampling), this will assist in assessing the sample representativeness.

The study could benefit from a more thorough literature review of existing research to compare findings and applicability.

The study would be better presented with a more detailed explanation of its sampling, data collection and analysis methodology for example inter-coder reliability. Also, while themes were provided clearly, the coding methods and thematic analysis should be provided explicitly.

The results are presented clearly but are mostly descriptive and lack adequate analysis especially regarding interrelating themes. These should be explored with more depth and presented rather than repeating all of the quotes.

To support validity, some results could benefit from statistical/evidence-based justifications.

Discussions identifies key barriers like health awareness and staffing constraints but should be more robust to include policy recommendations and scalability of the interventions. The discussions should provide some comparisons of results from other similar studies to highlight convergent or divergent themes and suggest some broader implications.

Biases in this study should be clearly stated to appreciate the limitations.

Practical and concise policy and practice recommendations, grounded in evidence would increase the impact of this study.

6. PLOS authors have the option to publish the peer review history of their article (what does this mean? ). If published, this will include your full peer review and any attached files.

**Do you want your identity to be public for this peer review?** For information about this choice, including consent withdrawal, please see our Privacy Policy .

Reviewer #1: No

---

## [Author Response · Author response to Decision Letter 1]

26 Feb 2025

Please see uploaded response to reviewers document.

---

## [Decision Letter · Decision Letter 1]

18 Mar 2025

Healthcare professionals' perspectives on barriers and facilitators to implementing a warning signs intervention for older rural-dwelling medical patients at risk for hospital readmission

PONE-D-24-37531R1

Dear Dr. Fox,

We’re pleased to inform you that your manuscript has been judged scientifically suitable for publication and will be formally accepted for publication once it meets all outstanding technical requirements.

Kind regards,

Mohd Ismail Ibrahim, MCom.Med

Academic Editor

PLOS ONE

Additional Editor Comments (optional):

Reviewers' comments:

Reviewer's Responses to Questions

**Comments to the Author**

1. If the authors have adequately addressed your comments raised in a previous round of review and you feel that this manuscript is now acceptable for publication, you may indicate that here to bypass the “Comments to the Author” section, enter your conflict of interest statement in the “Confidential to Editor” section, and submit your "Accept" recommendation.

Reviewer #2: All comments have been addressed

2. Is the manuscript technically sound, and do the data support the conclusions?

Reviewer #2: Yes

3. Has the statistical analysis been performed appropriately and rigorously? 

Reviewer #2: Yes

4. Have the authors made all data underlying the findings in their manuscript fully available?

Reviewer #2: Yes

5. Is the manuscript presented in an intelligible fashion and written in standard English?

Reviewer #2: Yes

6. Review Comments to the Author

Reviewer #2: This is a well written manuscript, it is methodologically sound and the findings are well presented.

7. PLOS authors have the option to publish the peer review history of their article (what does this mean? ). If published, this will include your full peer review and any attached files.

**Do you want your identity to be public for this peer review?** For information about this choice, including consent withdrawal, please see our Privacy Policy .

Reviewer #2: No

---

## [Editor Report · Acceptance letter]

PONE-D-24-37531R1

PLOS ONE

Dear Dr. Fox,

I'm pleased to inform you that your manuscript has been deemed suitable for publication in PLOS ONE. Congratulations! Your manuscript is now being handed over to our production team.

Kind regards,

on behalf of

Dr. PLOS Manuscript Reassignment

Staff Editor

PLOS ONE